

# Lipids associated with plant-bacteria interaction identified using a metabolomics approach in an *Arabidopsis thaliana* model

Jian-Bo Song[1,2], Rui-Ke Huang[1,2], Miao-Jie Guo[1,2], Qian Zhou[3], Rui Guo[1,2], Shu-Yuan Zhang[1,2], Jing-Wen Yao[1,2], Ya-Ni Bai[1,2] and Xuan Huang[1,2]

[1] College of Life Sciences, Northwest University, Shaanxi, Xi'an, China
[2] Key Laboratory of Resource Biology and Biotechnology in Western China (Ministry of Education), Provincial Key Laboratory of Biotechnology of Shaanxi, Shaanxi, Xi'an, China
[3] Shanghai Omicsspace Biotechnology Co.Ltd., Shanghai, Shanghai, China

## ABSTRACT

**Background**. Systemic acquired resistance (SAR) protects plants against a wide variety of pathogens. In recent decades, numerous studies have focused on the induction of SAR, but its molecular mechanisms remain largely unknown.

**Methods**. We used a metabolomics approach based on ultra-high-performance liquid chromatographic (UPLC) and mass spectrometric (MS) techniques to identify SAR-related lipid metabolites in an *Arabidopsis thaliana* model. Multiple statistical analyses were used to identify the differentially regulated metabolites.

**Results**. Numerous lipids were implicated as potential factors in both plant basal resistance and SAR; these include species of phosphatidic acid (PA), monogalactosyl-diacylglycerol (MGDG), phosphatidylcholine (PC), phosphatidylethanolamine (PE), and triacylglycerol (TG).

**Conclusions**. Our findings indicate that lipids accumulated in both local and systemic leaves, while other lipids only accumulated in local leaves or in systemic leaves. PA (16:0_18:2), PE (34:5) and PE (16:0_18:2) had higher levels in both local leaves inoculated with *Psm ES4326* or *Psm avrRpm1* and systemic leaves of the plants locally infected with *Psm avrRpm1* or *Psm ES4326*. PC (32:5) had high levels in leaves inoculated with *Psm ES4326*. Other differentially regulated metabolites, including PA (18:2_18:2), PA (16:0_18:3), PA (18:3_18:2), PE (16:0_18:3), PE (16:1_16:1), PE (34:4) and TGs showed higher levels in systemic leaves of the plants locally infected with *Psm avrRpm1* or *Psm ES4326*. These findings will help direct future studies on the molecular mechanisms of SAR.

# INTRODUCTION

A wide variety of pathogens have evolved to infect plants. Plant pathogens are divided into three categories on the basis of their infection mode: biotrophic, necrotrophic, and hemibiotrophic. Biotrophic pathogens infect specific types of plants to obtain nutrients

Corresponding author
Xuan Huang,
xuanhuang@nwu.edu.cn

required for growth (*Glazebrook, 2005*). Necrotrophic pathogens kill plants through the infection process and then obtain nutrients from the dead plant cells (*Mengiste, 2012*; *Weiberg et al., 2014*). Hemibiotrophic pathogens grow and multiply by infecting plant cells and then subsequently kill the plant. There are two types of plant immunity triggered by pathogens: pathogen-associated molecular pattern (PAMP)-triggered immunity (PTI) and effector-triggered immunity (ETI). PTI, the most basic type, is induced by pathogens entering through wounds or stomata (*Chisholm et al., 2006*). Conserved features of the pathogens are recognized by the plant pattern-recognition receptors (PRRs), triggering PTI through the accumulation of reactive oxygen species (ROS), increased mitogen-activated protein kinase (MAPK) signaling, and alteration of calcium ion ($Ca^{2+}$) concentration. These all lead to the initiation of antimicrobial proteins, accumulation of phytoalexins, as well as callose deposits (*Adigun et al., 2021*; *Nürnberger et al., 2004*). Hemibiotrophic pathogens such as *Pseudomonas syringae* have evolved type III secretion systems, whereby the pathogens inject effector molecules into the plant cytoplasm to suppress PTI (*Chang et al., 2005*; *Thomma, Nürnberger & Joosten, 2011*). In response to such suppression, resistance proteins (R proteins) of the nucleotide-binding site/leucine-rich repeat type (NBS-LRR) trigger ETI by acting as intracellular receptor proteins (*Dangl & Jones, 2001*). PTI and ETI often occur synergistically and both can trigger strong immune responses and systemic acquired resistance (SAR) (*Ngou et al., 2021*).

SAR is a defense mechanism that provides protection against a wide range of pathogens in distal uninoculated leaves (*Kachroo & Kachroo, 2020*; *Kachroo & Robin, 2013*). During the establishment period of SAR, some mobile signals from pathogen-infected tissues are transferred to distal tissues through the phloem, including glycerol-3-phosphate (G3P) (*Chanda et al., 2011*), azelaic acid (AzA) (*Jung et al., 2009*), pipecolic acid (*Návarová et al., 2012*), and *N*-hydroxypipecolic acid (NHP) (*Hartmann & Zeier, 2018*). Certain gaseous signaling molecules volatilized by plants (*e.g.*, terpenes and methylsalicylic acid) can act as mobile signals to induce resistance in adjacent plants (*Wang et al., 2014*). Besides these small molecules, several proteins related to signal transportation are involved in the activation of defense responses by distal tissues; for example, long distance transportation of the lipid transfer protein, DEFECTIVE IN INDUCED RESISTANCE1 (DIR1), plays a key role in systemic immunity (*Carella, Isaacs & Cameron, 2015*; *Champigny et al., 2013*; *Chanda et al., 2011*; *Wang et al., 2014*).

Metabolomics approaches based on ultra-high-performance liquid chromatography (UPLC-MS/MS) are increasingly being used for the elucidation of metabolites involved in the development, biotic and abiotic stress responses of plants. These methods allow for precise qualitative and quantitative analysis of primary and secondary compounds. *Chernova et al. (2018)* utilized UPLC-MS/MS to demonstrate that triacylglycerols play essential roles in cold tolerance mechanisms by comparing winter-type *vs.* spring-type plants. Ward et al. showed that the levels of phenolic, indolic, and amino acids in *Arabidopsis thaliana* change notably within 8 h of infection, based on UPLC-MS/MS data (*De Vos et al., 2007*). *Gao et al. (2020)* identified metabolites associated with systemic acquired resistance in *Arabidopsis* based on UPLC-MS/MS data. The plant defense responses to pathogenic microorganisms are often related to substantial modifications of the lipidome,

and several previous studies have measured *Arabidopsis* leaf lipids after *P. syringae* infection (*Kirik & Mudgett, 2009*; *Nandi, Welti & Shah, 2004*; *Nilsson et al., 2014*; *Vu et al., 2012*). In contrast to the significant advances in plant research using UPLC-MS/MS methods and a metabolomics approach, few studies have used a lipidomics approach for investigating SAR. In the present study, two phytopathogens were used to induce SAR, and SAR-related lipid metabolites were identified using UPLC-MS/MS-based lipidomics approaches. Various PAs, PCs, PEs, and TGs were identified in plant-bacterial interactions.

## MATERIALS & METHODS

### Plant materials

*A. thaliana* plants of Col-0 were used for our experiments. Seed dormancy was broken by treatment at 4 °C for 3 d, then the seeds were surface-sterilized in 10% NaClO for 10 min, rinsed three times with sterile water, and sown on MS solid medium (pH 5.8). Seedlings (four leaves) were transferred to pots containing a mixture of soil, perlite and vermiculite (8:3:1 v/v/v). Growth conditions included a relative humidity of 65% and a 16 h light (photon flux density 70 $\mu$mol m$^{-2}$ s$^{-1}$)/8 h dark cycle at 22 °C. Four-week-old plants that exhibited a uniform appearance were used for SAR experiments.

### Analyses of local and systemic resistance

*Arabidopsis* leaves were inoculated (by a syringe without needle) with avirulent *P. syringae pv. maculicola ES4326* harboring avrRpm1 (*Psm avrRpm1*) or virulent *P. syringae pv. maculicola ES4326* (*Psm ES4326*). The bacteria were grown at 28 °C in King's B medium containing the appropriate antibiotics (tetracycline 50 mg/L, streptomycin 50 mg/L) under permanent shaking (200 r/min, 28 °C). Overnight log-phase cultures were centrifuged at 3,000 rpm for 5 min, then the cultures were diluted with 10 mM MgSO$_4$ to OD$_{600}$ = 0.002. Three lower rosette leaves of 4-wk-old plants were infected with *Psm avrRpm1*; leaves treated with 10 mM MgSO$_4$ were used as control. On the 3rd day, three upper leaves of each plant were inoculated with *Psm ES4326*, the bacterial numbers were quantified at 3d post infiltration and phenotypes were compared at 4d post infiltration (*Li et al., 2020a*).

### Extraction of lipid metabolites

SAR was induced by inoculating three lower leaves of each plant with *Psm avrRpm1* or *Psm ES4326* suspension; leaves treated with 10 mM MgSO$_4$ were used as control (CK). After 48 h, leaves injected with MgSO$_4$, *Psm avrRpm1* or *Psm ES4326* (respectively termed LL-CK, LL-A, LL-V, leaves of developmental stages 4–6 as sample), and systemic leaves locally injected with MgSO$_4$-, *Psm avrRpm1*- or *Psm ES4326* (termed SL-CK, SL-A, SL-V, leaves of developmental stages 7–9 as sample) were collected to perform a LC-MS/MS and RT-qPCR analysis (*Gao et al., 2020*). Lipids were extracted according to the method as described previously (*Lu et al., 2019*; *Lu et al., 2018*). Freeze-dried samples (each 50 mg) were ground in liquid nitrogen, then one mL precooled extraction solution was added (CHCl$_3$/CH$_3$OH/300 mM ammonium bicarbonate, 30:41.5:3.5, v/v/v). The samples were then shaken at 100 rpm for 12 h at 4 °C and centrifuged at 1,000× g for 2 min at 4 °C. The supernatant was collected, 0.5 mL 1 M KCl was added, then the supernatant was

centrifuged at 1,000× g for 2 min at 4 °C. The upper aqueous phase was then removed and the sample was washed two times with one mL sterile water. After the upper aqueous phase was removed, the lower organic phase was dried under a stream of nitrogen. These samples were then dissolved in $(CH_3)_2CHOH/ C_2H_3N/ H_2O$ (3:3:1, v/v/v), centrifuged at 1,000× g for 2 min, and the supernatant was then used for a UPLC-MS/MS analysis. Quality control (QC) samples were mixed with equal amounts of each extraction solution.

## LC-MS/MS analysis

Samples were placed in an autosampler at 4 °C, and separated on a Dionex UltiMate-3000 with $C_{18}$ column: sample volume 3 μL, column temperature 45 °C, flow rate 0.3 mL/min; chromatographic mobile phase A: 0.1% formic acid/ acetonitrile (6:4, v/v); mobile phase B: acetonitrile/ propanol (1:9, v/v). The gradient elution program: 0−0.5 min, B maintained at 37%; 1.5–13.5 min, B increased from 37% to 70%; 13.5–15 min, B increased from 70% to 75%; 15.5–16.5 min, B increased linearly from 75% to 98%; 16.5–18 min, B maintained at 98%; 18-19 min, B decreased from 98% to 37%; 19-20 min, B maintained at 37%. The substances separated by UPLC were analyzed by Q-Exactive mass spectrometry. Samples were subjected to electrospray ionization (ESI) to yield positive and negative ions. The ESI source conditions were as follows: spray voltage (kV), 3.5 ESI+ and 3.2 ESI-; source temperature, 320 °C; sheath gas flow rate (Arb), 45; aux gas flow rate (Arb), 15; mass range (m/z), 150–2,000; full MS resolution, 70000; MS/MS resolution, 17500; TopN, 10; stepped NCE 15, 25, 35.

## Data analysis

Raw data were processed to ensure the retention time, peak alignment, and extraction of peak area were correct using the Lipidsearch 4.2.21 software program (*Blasco et al., 2017*; *Breitkopf et al., 2017*). To determine metabolite structures, the precision mass matching (25ppm), MS and MS/MS matching data were used to search the Lipid MAPS database. A multidimensional statistical analysis was performed using the SIMCA-P14.1 software (*Blasco et al., 2017*). Results were generated by an orthogonal partial least square discriminant analysis (OPLS-DA).

## Total RNA extraction and quantitative real-time PCR (RT-qPCR)

Total RNAs of the samples were isolated using a TRIzol reagent (Invitrogen), and the complementary DNA (cDNA) was synthesized using the PrimeScript First Strand cDNA Synthesis Kit (Takara). The 10 μL total reaction volume for RT-qPCR included 1 μL cDNA, 5 μL FastStart Essential DNA Green Master (Roche), 2 μL gene-specific primer, and 2 μL sterile water. The RT-qPCR program was as follows: denaturation at 95 °C for 10 min, 39 cycles of denaturation at 95 °C for 10 s and annealing at corresponding temperature for 30 s. Fold changes of various genes were calculated using the $2^{-\Delta\Delta Ct}$ method. AtACT8 was used as the reference gene. Primer sequences are listed in Table 1.

**Table 1 Primers used in RT-qPCR analyses of PR1 and PR5 genes related to SAR, and of lipid-related genes.**

| Gene | Sequences of primers (5′→3′) | Size of amplicon (bp) |
|---|---|---|
| MGD1 (AT4G31780) | F:GGAATGTATGGGTGCCTGTGACTG R: GCCTCTTGACCAGCGATGTAACC | 117 |
| GPAT1 (AT1G06520) | F:AGCCTGCACTTGGAATTGGAAGC R:TGCGTTGTTCTTGCTCATGGACTC | 113 |
| DGAT1 (AT2G19450) | F:TGGAAGAGGCGGCGGAGAAG R:TGAAGATTGCGTCGGAGCTAAGTG | 116 |
| AAPT1 (AT1G13560) | F:TGATGGGAAGCAAGCAAGAAGGA R:CACAAGCAAGCGCGTCACAAC | 83 |
| CCT1 (AT2G32260) | F:CGCACTGAAGACGGCCTTTCC R:TCGTAGATCCCATCGGCGTAGAC | 110 |
| PECT1 (AT2G38670) | F:CTTCGTCAAGCTCGTGCTCTCG R:AACTTCATCCACCCACTTCACAGC | 144 |
| PR1 (AT2G14610) | F: CTCTTGTAGGTGCTCTTGTTC R: CCTCTTAGTTGTTCTGCGTAG | 160 |
| PR5 (AT1G75040) | F:AAGAGTGCCTGTGAGAGGTT R:TTCGTCGTCATAAGCGTAGC | 144 |
| ACT8 (AT1G49240) | F: TGTGCCTATCTACGAGGGTTT R: TTTCCCGTTCTGCTGTTGT | 137 |

## RESULTS

### Systemic acquired resistance (SAR) experiments

The quality of the samples was assessed using the RT-qPCR measurement of accumulation levels of both *PR1* and *PR5* transcripts in the systemic leaves locally injected with $MgSO_4$-, *Psm avrRpm1*- or *Psm ES4326* (SL-CK, SL-A, SL-V). *PR1* and *PR5* levels were higher in the systemic leaves of the plants locally infected with *Psm avrRpm1* or *Psm ES4326* compared to the systemic leaves of the locally $MgSO_4$ treated plant (Fig. 1A). Successful SAR induction was confirmed by performing secondary infection with *Psm ES4326* on the systemic leaves of plants locally injected with $MgSO_4$, *Psm avrRpm1* or *Psm ES4326*. Phenotypes were compared after 4 d. Compared with $MgSO_4$ treated plants, the systemic leaves of plants locally infected with *Psm avrRpm1* or *Psm ES4326* had a lower degree of chlorosis and a smaller necrosis area (Fig. 1C). The number of bacteria in the systemic leaves of the plants locally infected with *Psm avrRpm1* or *Psm ES4326* were significantly lower than in $MgSO_4$-treated leaves (Fig. 1B). These findings confirm successful SAR induction by *Psm avrRpm1* or *Psm ES4326*. These samples were subsequently used for the lipid metabolomics analysis.

### OPLS-DA

OPLS-DA accurately reveals intrinsic differences by removing data unrelated to classification, thus improving analytical precision. We obtained values of 0.996 for $R^2Y$ (which reflects interpretive ability of variable Y) and 0.994 for $Q^2$ (which reflects predictive ability of the model), indicating that the OPLS-DA reliably explains the differences between

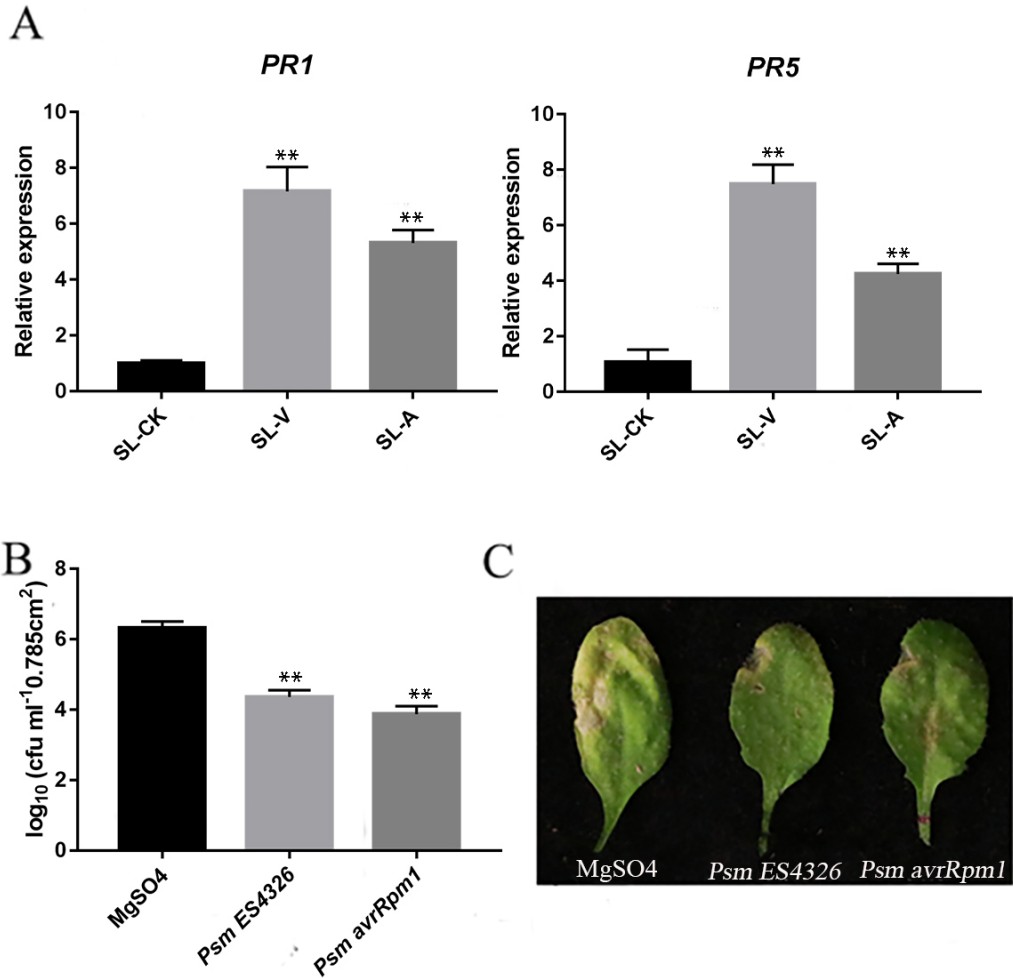

**Figure 1** **Systemic acquired resistance (SAR) experiments.** (A) Expression analysis of *PR1* and *PR5* in systemic leaves of plants inoculated locally with *Psm avrRpm1* or *Psm ES4326* in comparison with $MgSO_4$-treated plants. (B) Growth of virulent *Psm ES4326* on systemic leaves of Col-0. Three lower rosette leaves of 4-wk-old plants of Col-0 genotype were injected with $MgSO_4$, *Psm avrRpm1*; 2 d later, upper systemic leaves were injected with *Psm ES4326*. At 3 d, one leaf disc (diameter five mm) was taken from each infiltrated systemic leaf. (C) Phenotypes resulting from secondary infection with *Psm ES4326* on distal leaves of plants locally injected with *Psm avrRpm1*, *Psm ES4326*, or $MgSO_4$. V: *Psm ES4326*; A: *Psm avrRpm1*; CK: $MgSO_4$; SL-A: systemic leaves of locally *Psm avrRpm1* inoculated plants; SL-V: systemic leaves of locally *Psm ES4326* inoculated plants; SL-CK: systemic leaves of locally $MgSO_4$ inoculated plants. Statistical significance was determined by *t*-test ($P < 0.05$). Asterisks (**) represent $P < 0.01$.

the treatment and control groups (Fig. 2). The long distance between the treatment groups and control group indicates that the systemic leaves of the plants locally infected with *Psm avrRpm1* or *Psm ES4326* produce some metabolites that are different from those in the systemic leaves of the locally $MgSO_4$ treated plants (Fig. 2). The short distance between the systemic leaves of the plants locally infected with *Psm avrRpm1* or *Psm ES4326* indicates that *Psm avrRpm1* and *Psm ES4326* induce similar metabolites in systemic leaves (Fig. 2).

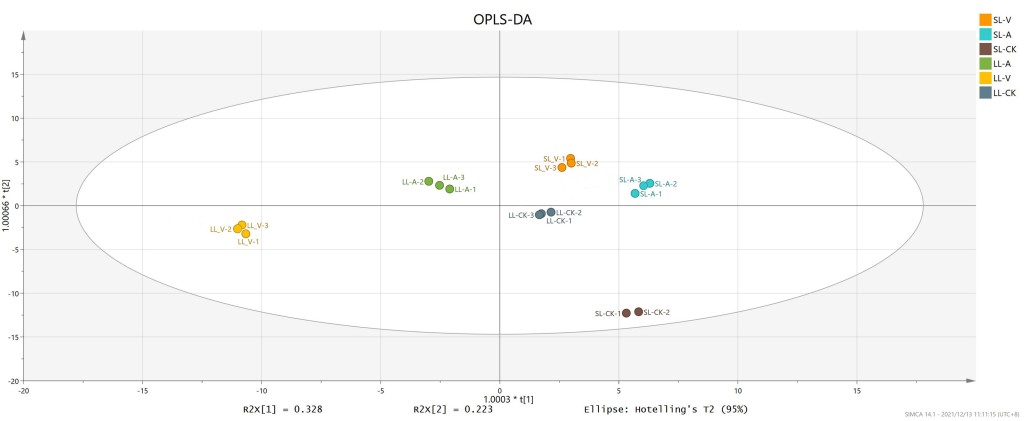

**Figure 2** OPLS-DA score plots of leaves injected with MgSO$_4$, *Psm avrRpm1*, or *Psm ES4326* and distal leaves of plants locally injected with MgSO$_4$, *Psm avrRpm1*, or *Psm ES4326*. OPLS-DA score plots were performed to reveal intrinsic difference within the signals of different groups. *X*-axis is the correlation coefficient between permuted and original response variables, which represents the degree of randomization of response variable y. SL-A: systemic leaves of locally *Psm avrRpm1* inoculated plants; SL-V: systemic leaves of locally *Psm ES4326* inoculated plants; SL-CK: systemic leaves of locally MgSO$_4$ treated plants (SL-CK groups were two biological replicates).

## Identification of differentially regulated metabolites (DRMs) associated with SAR

A total of 127 lipid metabolites were identified in infected leaves and distal uninoculated leaves (File S1). A heatmap analysis of the DRMs in the samples is shown in Fig. 3. We found from the heat map that PE (34:5), PE (34:4), PA (16:0_18:3), PE (16:1_16:1) accumulated in the systemic leaves of the plants locally infected with *Psm avrRpm1* or *Psm ES4326*. PE (16:0_18:2) and PE (34:5) accumulated in the local leaves infected with *Psm ES4326*. The screening of DRMs is shown in Fig. 4A (fold change < 0.83 or > 1.2; VIP > 1; *p* < 0.05). There were 10 DRMs with increased abundance and 15 DRMs with decreased abundance in leaves inoculated with *Psm ES4326* (LL-V *vs.* LL-CK), including MGDG (16:1_18:2), MGDG (18:2_16:2), PA (18:2_18:2), PE (16:0_18:3), PC (36:2) and PE (34:5). There were 15 DRMs with decreased abundance in leaves inoculated with *Psm avrRpm1* (LL-A *vs.* LL-CK), including PC (36:3), MGDG (18:2_16:2), and PE (34:4). The DRMs with either increased or decreased levels in infected leaves are presumably related to the plant-bacteria interaction. There were 19 DRMs in the systemic leaves of the locally *Psm avrRpm1*-infected plant (SL-A *vs.* SL-CK) and 30 DRMs in the systemic leaves of the locally *Psm ES4326*-infected plant (SL-V *vs.* SL-CK) that showed significant changes, including PE (34:5), PA (16:0_18:3) and PE (16:1_16:1). Some DRMs were identified in both the infected leaves and distal uninoculated leaves. A Venn diagram analysis showed 8 DRMs commonly found in the leaves of plants inoculated with *Psm avrRpm1* or *Psm ES4326* (LL-A *vs.* LL-CK and LL-V *vs.* LL-CK) and 9 DRMs commonly found in the systemic leaves of the plants locally infected with *Psm avrRpm1* or *Psm ES4326* (SL-A *vs.* LL-CK and SL-V *vs.* LL-CK) (Fig. 4B). Because *Psm avrRpm1* and *Psm ES4326* induce very similar SAR in

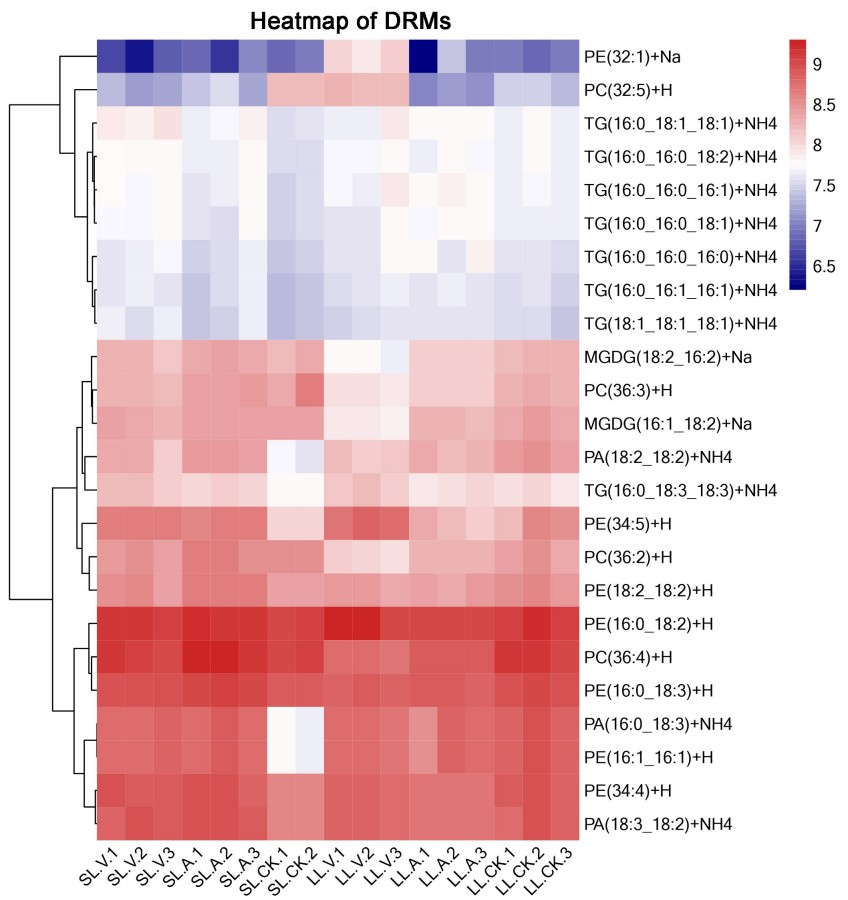

**Figure 3** Clustering analysis of identified metabolites in leaves injected with MgSO₄, *Psm avrRpm1*, or *Psm ES4326* and distal leaves of plants locally injected with MgSO₄, *Psm avrRpm1*, or *PsmES4326*. Each square represents a metabolite; the color scale on the right represents the abundance of lipids (red represents increased abundance, blue represents decreased abundance). The phylogeny lines on the left represent the clustering analysis results of metabolites. The abundance values were calculated by log₁₀ (SL-CK groups were two biological replicates).

Col-0, there were common DRMs induced by *Psm avrRpm1* and *Psm ES4326* in systemic leaves, such as PE (34:5), PE (34:4), PA (16:0_18:3), and PE (16:1_16:1) (Table 2).

## Quantitative analysis of expression of PA-, PC-, PE-, TG-, and MGDG-related genes

To explore the expression level of genes related to lipid biosynthesis, we performed an RT-qPCR analysis of expression levels in the bacterium-treated and distal uninoculated leaves of six genes involved in the synthesis of PAs, MGDGs, PCs, PEs, and TGs (Fig. 5). The genes were selected based on their functions in the specific metabolic pathways that we identified. PA, produced by the sequential acylation of G3P by acyltransferase (GPAT1), is the substrate for CDP-DG synthesis. Monogalactosyldiacylglycerol synthase 1 (MGD1) is a key enzyme in MGDG synthesis. *PECT1* (encodes a mitochondrial ethanolamine-phosphate cytidylyltransferase) and *CCT1* (encodes a phosphorylcholine cytidylyltransferase) are

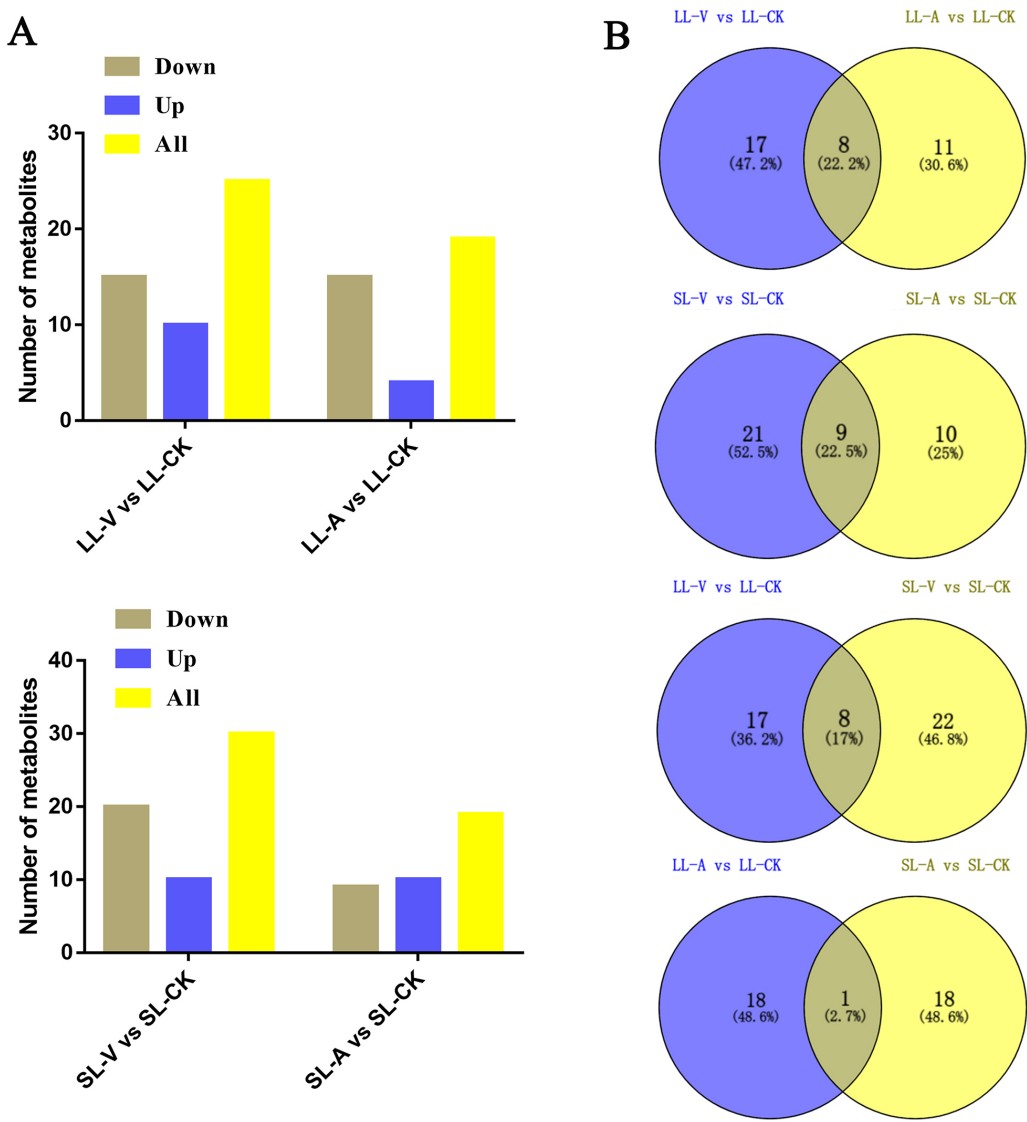

**Figure 4 Responses of *Arabidopsis* metabolites to *Psm avrRpm1* and *Psm ES4326* infection in local leaves and distal leaves of plants inoculated locally with *Psm avrRpm1* or *Psm ES4326*.** (A) Numbers of commonly increased or decreased levels of metabolites in leaves infected with *Psm avrRpm1* or *Psm ES4326* (LL-V *vs.* LL-CK, LL-A *vs.* LL-CK) and distal leaves of plants locally infected with *Psm avrRpm1* or *Psm ES4326* in comparison with MgSO₄-treated plants (SL-V *vs.* SL-CK, SL-A *vs.* SL-CK). (B) Venn diagrams illustrating common metabolites in leaves infected with *Psm ES4326* or *Psm avrRpm1* (LL-V *vs.* LL-CK and LL-A *vs.* LL-CK), distal leaves of plants locally infected with *Psm ES4326* or *Psm avrRpm1* (SL-V *vs.* SL-CK and SL-A *vs.* SL-CK), leaves infected with *Psm ES4326* or distal leaves of locally *Psm ES4326*-infected plants (LL-V *vs.* LL-CK and SL-V *vs.* SL-CK), leaves infected with *Psm avrRpm1* and distal leaves of locally *Psm avrRpm1*-infected plants (LL-A *vs.* LL-CK and SL-A *vs.* SL-CK). V: *Psm ES4326*; A: *Psm avrRpm1*; CK: MgSO₄; SL-A: systemic leaves of locally *Psm avrRpm1* inoculated plants; SL-V: systemic leaves of locally *Psm ES4326* inoculated plants; SL-CK: systemic leaves of locally MgSO₄ inoculated plants.

**Table 2    Differentially regulated metabolites (DRMs) in LL-A *vs.* LL-CK, LL-V *vs.* LL-CK, SL-A *vs.* SL-CK, and SL *vs.* SL-CK.**

| DLM | Formula | Fold change and *p*-value | | | | |
| --- | --- | --- | --- | --- | --- | --- |
| | | LL-V *vs.* LL-CK | LL-A *vs.* LL-CK | SL-V *vs.* SL-CK | SL-A *vs.* SL-CK (min) | RT |
| MGDG (16:1_18:2) | C43 H76 O10 | 0.32(1.37E−02) | 0.73(3.76E−02) | | 0.45(8.02E−03) | 9.69 |
| MGDG (16:2_18:2) | C43 H74 O10 | 0.30(2.90E−04) | 0.65(2.26E−03) | | | 9.43 |
| PA (18:2_18:2) | C39 H73 O8 N1 P1 | 0.50(4.86E−03) | 0.67(2.25E−02) | 4.21(4.96E−02) | 6.19(1.47E−03) | 7.70 |
| PA (16:0_18:3) | C37 H71 O8 N1 P1 | 0.79(1.05E−01) | | 11.89(6.35E−05) | 11.88(5.67E−03) | 8.97 |
| PA (16:0_18:2) | C37 H73 O8 N1 P1 | 2.07(3.28E−01) | 0.54(4.46E−03) | | 1.53(4.56E−01) | 8.98 |
| PA (18:3_18:2) | C39 H71 O8 N1 P1 | | 0.69(3.18E−02) | 1.93(1.92E−02) | 2.30(3.33E−03) | 7.83 |
| PC (36:2) | C44 H85 O8 N1 P1 | 0.39(4.08E−03) | 0.67(3.39E−02) | | | 10.45 |
| PC (36:3) | C44 H83 O8 N1 P1 | 0.44(2.09E−04) | 0.64(1.77E−03) | | | 9.89 |
| PC (36:4) | C44 H81 O8 N1 P1 | 0.64(3.17E−03) | 0.64(8.58E−03) | | 1.54(5.4E−02) | 9.16 |
| PC (32:5) | C40 H71 O8 N1 P1 | 6.70(3.27E−05) | | 0.11(1.97E−05) | 0.16(2.84E−4) | 7.21 |
| PE (16:0_18:3) | C39 H73 O8 N1 P1 | 0.79(3.94E−02) | 0.83(4.32E−2) | 1.23(8.44E−04) | 1.43(6.87E−02) | 8.95 |
| PE (34:5) | C39 H69 O8 N1 P1 | 1.95(2.30E−02) | | 3.82(1.14E−05) | 3.74(2.31E−04) | 8.34 |
| PE (16:1_16:1) | C37 H71 O8 N1 P1 | 0.79(1.05E−01) | | 11.89(6.35E−05) | 11.88(5.57E−03) | 9.01 |
| PE (16:0_18:2) | C39 H75 O8 N1 P1 | 1.51(4.24E−01) | | 1.19(1.35E−02) | 1.36(1.01E−01) | 10.02 |
| PE (34:4) | C39 H71 O8 N1 P1 | | 0.65(2.33E−03) | 2.08(1.67E−03) | 2.23(1.18E−02) | 8.33 |
| PE (18:2_18:2) | C41 H75 O8 N1 P1 | | | 1.24(2.41E−01) | 1.73(3.48E−04) | 8.50 |
| TG (16:0_16:0_16:0) | C51 H102 O6 N1 | | | 1.70(2.77E−02) | 1.32(1.92E−01) | 15.14 |
| TG (16:0_16:0_16:1) | C51 H100 O6 N1 | | | 1.83(2.03E−02) | 1.47(2.14E−01) | 14.74 |
| TG (16:0_16:1_16:1) | C51 H98 O6 N1 | | | 1.73(6.22E−03) | 1.47(2.83E−01) | 14.53 |
| TG (16:0_16:0_18:1) | C53 H104 O6 N1 | | | 1.75(1.45E−02) | | 15.06 |
| TG (16:0_16:0_18:2) | C53 H102 O6 N1 | | | 1.78(2.36E−03) | | 14.82 |
| TG (16:0_18:1_18:1) | C55 H106 O6 N1 | | | 2.16(6.11E−03) | | 15.05 |
| TG (16:0_18:3_18:3) | C55 H98 O6 N1 | | | 2.50(7.76E−03) | 1.90(6.16E−03) | 13.89 |
| TG (18:1_18:1_18:1) | C57 H108 O6 N1 | | | 1.71(2.17E−02) | | 15.1 |

essential genes in PE and PC synthesis. Amino-alcohol phosphotransferase (AAPT1) simultaneously catalyzes the synthesis of PC and PE. TG is synthesized by diacylglycerol acyltransferase (DGAT1), using acyl-CoA as the acyl donor. We found that the expression level of *DGAT1* was significantly higher in leaves inoculated with *Psm avrRpm1* compared to leaves treated with $MgSO_4$, and that the *CCT1* transcript level was upregulated significantly in leaves inoculated with *Psm ES4326* (Fig. 5). *PECT1* expression levels did not differ significantly among the local leaf groups, but were higher in the systemic leaves of the plants locally infected with *Psm avrRpm1* or *Psm ES4326* than in the systemic leaves of the locally $MgSO_4$ treated plant. This study found that the expression level of *GPAT1* was induced by both *Psm avrRpm1* and *Psm ES4326* in both inoculated and systemic leaves.

## DISCUSSION

Systemic acquired resistance (SAR) provides defense protection against a wide range of pathogens in whole plants following primary inoculation (*Maldonado et al., 2002*). During the past decade, certain metabolites have been studied extensively at the transcriptional and metabolic levels, but still little is known about lipid responses to *Psm* (*Gruner et*

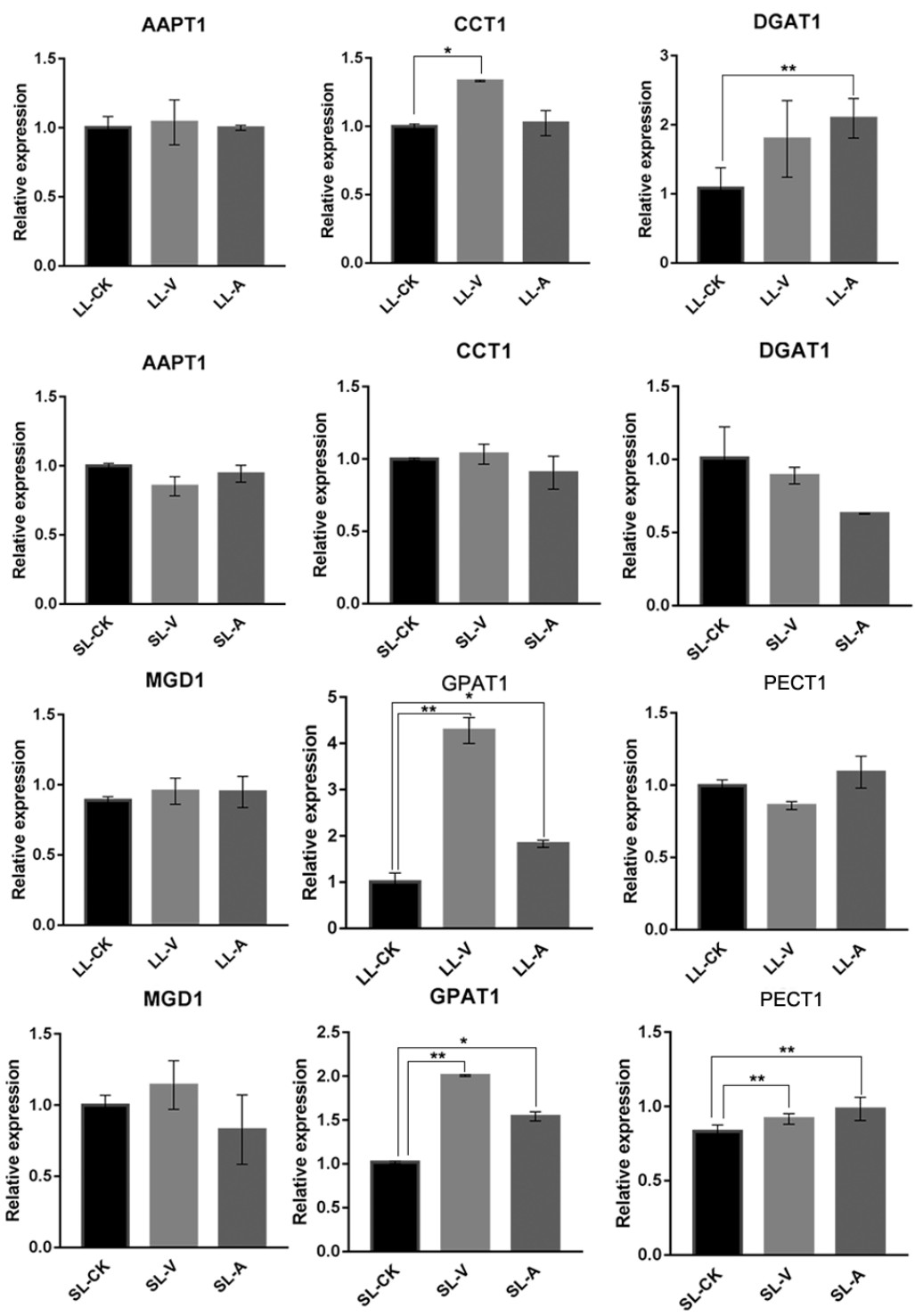

**Figure 5** Expression analyses of genes related to lipids biosynthesis by RT-qPCR following treatments with *Psm avrRpm1* and MgSO$_4$. The leaves injected with MgSO$_4$, *Psm avrRpm1* or *Psm ES4326* (LL-CK, LL-A, LL-V) and distal leaves of plants locally injected with MgSO$_4$, *Psm avrRpm1*, or *Psm ES4326* (SL-CK, SL-A, and SL-V) were collected at 48 h. (continued on next page...)

**Figure 5 (...continued)**
Relative transcript abundance of *AAPT1, CCT1, DGAT1, MGD1, GPAT1* and *PECT1* were determined in local and distal leaves of Col-0 by RT-qPCR. The RT-qPCR analysis has 3 biological replicates for each treatment. Statistical significance was determined by one-way ANOVA ($P < 0.05$). * represents $P < 0.05$, Asterisks (**) represent $P < 0.01$. V: *Psm ES4326*; A: *Psm avrRpm1*; CK: $MgSO_4$; SL-A: systemic leaves of locally *Psm avrRpm1* inoculated plants; SL-V: systemic leaves of locally *Psm ES4326* inoculated plants; SL-CK: systemic leaves of locally $MgSO_4$ inoculated plants.

*al., 2013*; *Schwachtje et al., 2018*). *Attaran, Rostás & Zeier (2008)* found that plants can induce the accumulation of the terpenoids (E,E)-4,8,12-trimethyl-1,3,7,11-tridecatetraene (TMTT), β-ionone and α-farnesene after inoculation of virulent or avirulent *P. syringae* strains; and they proved that pathogen-induced synthesis of TMTT is controlled through jasmonic acid (JA)-dependent signaling, but is independent of a functional salicylic acid (SA) pathway. *Griebel & Zeier (2010)* found that the accumulation of phytosterol stigmasterol is a significant part of the plant metabolic process that occurs upon bacterial leaf infection. *Stahl et al. (2019)* show that inoculation of *Arabidopsis* leaves with the bacterial pathogen *Pseudomonas syringae* induces the expression of genes involved in the early steps of tocopherol biosynthesis and triggers a strong accumulation of γ-tocopherol, a moderate production of δ-tocopherol, and the generation of the benzoquinol precursors of tocopherols. The connection between arabidopsides and *Pseudomonas syringae* has also been established in previous studies. *Andersson et al. (2006)* found that the levels of arabidopsides were highest 4 h after local leaves were infected by *P. syringae* but found no significant differences in the changes of nonoxidized fatty acids during the experiments. These SAR-related lipid derivatives were not identified in this study likely because of the varying sample collection times and the methods used for lipid extraction. The previous methods for extracting SAR-related metabolites were aimed at extracting metabolites rather than targeting lipids. We used a metabolomics approach based on high-resolution mass spectrometry to identify new lipid metabolites related to the plant-pathogen interaction in both local and systemic leaves. A multivariate statistical analysis revealed significant metabolomic differences between *Arabidopsis* plants injected with *Psm* and those injected with $MgSO_4$. SAR-related lipids were identified for the first time based on a LC-MS/MS-based lipidomics approach, and numerous DRMs were identified in the plant-pathogen interaction; these included MGDG $(16:1\_18:2)_{FC(SL-V)=0.45}$, PA $(18:2\_18:2)_{FC(SL-V)=4.21}$, PC $(32:5)_{FC(SL-V)=0.11}$, PE $(16:1\_16:1)_{FC(SL-V)=11.89}$ and TG $(16:0\_18:3\_18:3)_{FC(SL-V)=2.5}$. We also performed a correlation analysis of DRMs (File S2). The results showed that some lipids have different correlations between local leaves and systemic leaves. For example, there is a positive correlation between PE $(34:5)_{FC(LL-V)=1.95,FC(SL-V)=3.82}$ and PA $(18:3\_18:3)_{FC(LL-V)=1.67,FC(SL-V)=7.36}$ in local and systemic leaves. PC $(32:5)_{FC(SL-V)=0.11}$ and PE $(34:5)_{FC(SL-V)=3.82}$ are negatively correlated in systemic leaves, but positively correlated in local leaves.

## Phosphatidic acid (PA)

We observed increased levels of PA $(18:2\_18:2)_{FC(SL-A)=6.19,FC(SL-V)=4.21}$ and PA $(16:0\_18:3)_{FC(SL-V)=11.89,FC(SL-A)=11.88}$ in the systemic leaves of plants locally infected

with *Psm avrRpm1* or *Psm ES4326*, reflecting the role of PAs in biotic stress (Table 2). PAs are important plant lipids involved in regulating physiological processes (*e.g.*, protein phosphorylation, cell proliferation and growth) and cell membrane composition. They also function as signaling molecules in various metabolic pathways in response to biotic and abiotic stresses (*Bargmann & Munnik, 2006*; *Wang et al., 2006*). Under various stress conditions in *A. thaliana*, phospholipase D (PLD) hydrolyzes the phosphodiester bond at the phospholipid terminal to produce defense-signaling molecules such as second-messenger PA (*Adigun et al., 2021*). PLD promotes phosphorylation of downstream signaling factors by regulating G proteins, and helps convert extracellular signals into intracellular signals (*Zhao et al., 2007*). PA levels are upregulated at the wound site and in the surrounding undamaged areas in mechanically wounded castor bean leaves, and systemically upregulated in soybean seedlings (*Lee, Hirt & Lee, 2001*; *Ryu & Wang, 1996*). PAs produced through PLD catalysis play essential roles in plant-pathogen interactions, and PA levels are correlated with PLD transcription levels (*Wang et al., 2006*). *De Torres Zabela et al. (2002)* observed that PLD transcription levels were upregulated in *A. thaliana* leaves infected by both virulent and avirulent strains of *P. syringae pv. Tomato*.

## Monogalactosyldiacylglycerol (MGDG)

In this work, we observed decreased levels of MGDG $(16:1\_18:2)_{FC(LL-V)=0.32,FC(LL-A)=0.73}$ and MGDG $(16:2\_18:2)_{FC(LL-V)=0.30,FC(LL-A)=0.65}$ in both leaves inoculated with *Psm ES4326* or *Psm avrRpm1*, indicating the importance of these lipid metabolites in the plant-pathogen interaction (Table 2). Galactolipids comprise ∼80% of membrane lipids in plants (*Kelly & Dörmann, 2004*). MGDG, a unique galactolipid localized in the chloroplast thylakoid membrane, is synthesized by MGD1, which transfers a galactose residue to diacylglycerol (*Awai et al., 2001*). Digalactosyldiacylglycerol (DGDG) is subsequently synthesized by digalactosyldiacylglycerol synthase1 (DGD1) catalysis (*Dörmann, Balbo & Benning, 1999*; *Froehlich, Benning & Dörmann, 2001*). MGDG and DGDG levels are altered in the chloroplast thylakoid membranes through exposure to abiotic stress (*Du et al., 2018*; *Li et al., 2020b*). *Dörmann et al. (1995)* reported that these compounds play key roles in the remodeling of membrane lipids under freezing stress. They are both associated with SAR. Gao et al. found that they regulate SAR in differing ways and function nonredundantly (*Gao et al., 2014*). DGDG is involved in nitric oxide (NO) synthesis during SAR, whereas MGDG regulates the synthesis of AzA, which functions downstream of NO. The α-Gal-β-Gal head of DGDG is essential for establishing SAR; DGDG with a β-Glc-β-Gal head is unable to perform this function.

## Phosphatidylcholine (PC)

In the present study, all detected PCs showed significant changes in LL-V relative to LL-CK, including PC $(36:2)_{FC(LL-V)=0.39}$, PC $(36:3)_{FC(LL-V)=0.44}$ and PC $(36:4)_{FC(LL-V)=0.64}$ (Table 2). The different levels of PCs in leaves inoculated with *Psm ES4326* were detected in leaves inoculated with *Psm avrRpm1*; however, the levels of all PCs in leaves inoculated with *Psm avrRpm1* were lower than in leaves treated with MgSO$_4$. A possible explanation is that the differing virulence factor (*AvrRpm1*) secreted by T3SS in *Psm avrRpm1* and *Psm ES4326*

result in differing PC accumulation levels. PC is a major, fundamental membrane lipid in eukaryotes, but much less has been reported on its role in prokaryotes (*Aktas et al., 2010*). Because it is not detectable in the membranes of *Escherichia coli* and *Bacillus subtilis*, two well-studied model species, its function in bacteria was not appreciated for many decades. However, studies of bacterial genome sequences around 2003 revealed that PC is present in 10% of bacterial species (*Sohlenkamp, López-Lara & Geiger, 2003*). Subsequent studies of PC were focused on host-pathogen interactions (*Comerci et al., 2006*; *Conde-Alvarez et al., 2006*). *Xiong et al. (2014)* observed that *P. syringae* strains defective in PC synthesis lost virulence for infecting plants, and suggested that the type III secretion system (T3SS) is directly dependent on PC for full virulence.

## Phosphatidylethanolamine (PE)

We observed significantly increased levels of PEs, including PE(34:5)$_{FC(SL-V)=3.82,FC(SL-A)=3.74}$ and PE (16:1_16:1)$_{FC(SL-V)=11.89,FC(SL-A)=11.88}$ in the systemic leaves of locally SAR-induced plants, suggesting involvement of these PEs in SAR establishment (Table 2). PE is an abundant membrane phospholipid in both *Arabidopsis* and bacteria (*Randle, Albro & Dittmer, 1969*). Many bacteria are able to split PE into phosphate and ethanolamine as a source of carbon and nitrogen (*Blackwell, Scarlett & Turner, 1976*; *Proulx & Fung, 1969*). The ethanolamine utilization operon (eut) encodes ethanolamine ammonialyase, which breaks down ethanolamine into acetaldehyde and ammonia (*Bradbeer, 1965a*; *Bradbeer, 1965b*). Utilization of ethanolamine in host-pathogen interactions is evidenced by the upregulation of eut genes. *Münch et al. (2008)* observed upregulation of eutABC in the hemolymph of *Galleria mellonella* (greater wax moth) infected by *Photorhabdus luminescens*, suggesting a key role of ethanolamine during this host-pathogen interaction. The upregulation of eut genes has also been reported in plant-pathogen interactions. *Yang et al. (2004)* found that eutR expression was induced by plant pathogen *Erwinia chrysanthemi* in spinach, and that a mutant eutR strain caused localized maceration of African violets, but did not cause systemic infection. These findings indicate that ethanolamine plays important roles in both pathogens and host plants.

## Triacylglycerol (TG)

After the leaves were inoculated with *Psm ES4326* bacteria, we observed increased levels of TGs in the systemic leaves of the locally *Psm ES4326*-infected plant, including TG (16:0_16:0_16:0)$_{FC(SL-V)=1.70}$, TG (16:0_16:0_16:1)$_{FC(SL-V)=1.83}$, TG (16:0_18:1_18:1)$_{FC(SL-V)=2.16}$ and TG (16:0_18:3_18:3)$_{FC(SL-V)=2.50}$. This result suggests that TGs may be important in this stage of SAR (Table 2). Triacylglycerol accounts for ~15% of the dry weight of plant leaves (*Sanjaya Durrett, Weise & Benning, 2011*; *Winichayakul et al., 2013*). It is a major stored energy source during seed germination and also an important factor in adult plant development (*Fan, Yan & Xu, 2013*; *Graham, 2008*).Triacylglycerol is involved in a variety of abiotic stresses, including desiccation tolerance and freezing tolerance (*Gasulla et al., 2013*; *Moellering, Muthan & Benning, 2010*). Under nitrogen deprivation conditions, both plant vegetative tissues and seeds show an increased expression of both DGAT1 (diacylglycerol acyltransferase) and PDAT

(phospholipid:diacylglycerol acyltransferase), the key enzyme for triacylglycerol synthesis (*Lee et al., 2018*). Triacylglycerol accumulation has a clear, protective effect against cell death in plants (*Fan, Yu & Xu, 2017*). Studies of biotic stress have shown that increased levels of unusual fatty acids in triacylglycerol protect *Arabidopsis* against predation by newly-hatched cabbage looper caterpillars (*Tunaru et al., 2012*).

## CONCLUSIONS

Lipids are involved in a vast array of developmental processes and various stress responses in plants. During bacterial infection of plants, plants resist pathogen attacks through innate immune responses, which are initiated by cell surface-localized pattern-recognition receptors (PRRs) and intracellular nucleotide-binding domain leucine-rich repeat containing receptors (NLRs) leading to PTI and ETI. SAR is then initiated to enhance resistance to pathogenic bacteria in the systemic leaves of plants locally infected with *Psm avrRpm1* or *Psm ES4326*. In the present study, we used molecular biology and metabolomics approaches to investigate differentially regulated metabolites (DRMs) during plant-pathogen interaction. Among the 127 identified metabolites, 17 and 22 DRMs showed significant changes in local leaves and systemic leaves, respectively (Table 2). PC (32:5) were displayed in high abundance in leaves inoculated with *Psm ES4326*, indicating these DRMs were induced in local leaves infected with *P. syringae*. Other DRMs, including PA (18:2_18:2), PA (16:0_18:3), PA (18:3_18:2), PE (16:0_18:3), PE (16:1_16:1), PE (34:4) and TGs were displayed in high abundance in systemic leaves of plants locally infected with *Psm avrRpm1* or *Psm ES4326*. PA (16:0_18:2), PE (34:5) and PE (16:0_18:2) were displayed in high levels in both local leaves inoculated with *Psm ES4326* or *Psm avrRpm1* and systemic leaves of the plants locally infected with *Psm avrRpm1* or *Psm ES4326*.

### Funding

This study was supported by grants from the National Natural Science Foundation of China (31300223), the Major Project of Basic Research Program of Natural Sciences of Shaanxi Province (2021JZ-41), Natural Science Foundation of Shaanxi Province (2016JM3001), Opening Foundation of Key Laboratory of Resource Biology and Biotechnology in Western China (Northwest University), Ministry of Education, First-class University and Academic programs of Northwest University, Northwest University Graduate Innovation and Creativity Funds (YZZ17152), and National Training Programs of Innovation and Entrepreneurship for Undergraduate (201910697021). The funders had no role in study design, data collection and analysis, decision to publish, or preparation of the manuscript.

### Grant Disclosures

The following grant information was disclosed by the authors:
National Natural Science Foundation of China: 31300223.
Major Project of Basic Research Program of Natural Sciences of Shaanxi Province: 2021JZ-41.

Natural Science Foundation of Shaanxi Province: 2016JM3001.
Opening Foundation of Key Laboratory of Resource Biology and Biotechnology in Western China.
Ministry of Education, First-class University and Academic programs of Northwest University.
Northwest University Graduate Innovation and Creativity Funds: YZZ17152.
National Training Programs of Innovation and Entrepreneurship for Undergraduate: 201910697021.

## Competing Interests

The authors declare there are no competing interests.

## Author Contributions

- Jian-Bo Song conceived and designed the experiments, performed the experiments, prepared figures and/or tables, authored or reviewed drafts of the paper, and approved the final draft.
- Rui-Ke Huang conceived and designed the experiments, performed the experiments, prepared figures and/or tables, and approved the final draft.
- Miao-Jie Guo and Qian Zhou conceived and designed the experiments, performed the experiments, analyzed the data, prepared figures and/or tables, authored or reviewed drafts of the paper, and approved the final draft.
- Rui Guo and Shu-Yuan Zhang conceived and designed the experiments, analyzed the data, prepared figures and/or tables, and approved the final draft.
- Jing-Wen Yao and Xuan Huang conceived and designed the experiments, analyzed the data, authored or reviewed drafts of the paper, and approved the final draft.
- Ya-Ni Bai analyzed the data, authored or reviewed drafts of the paper, and approved the final draft.

## Data Availability

The raw measurements are available in the Supplementary Files.

The mass spectrometry proteomics data are available at the ProteomeXchange Consortium via the iProX partner repository: PXD027990.

## Supplemental Information

Supplemental information for this article can be found online at http://dx.doi.org/10.7717/peerj.13293#supplemental-information.

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
