# Peer review of "Lipids associated with plant-bacteria interaction identified using a metabolomics approach in an Arabidopsis thaliana model"

_PeerJ, doi:10.7717/peerj.13293_

## Round 0.1 · original submission · Major Revisions

All reviewers agree that your study represents an important contribution to the field. Specifically, please focus on a careful revision of the description of the method section and explain in detail how you identified the described lipid molecular species (reviewer #2).

Reviewer 1 ·

Basic reporting

The manuscript needs language editing. The quality now is quite uneven. Some parts are good, whereas some others (methods in particular) are hard to follow. There are many abbriviations not commonly used in the pland pathogen interactions field. I would recommend using more commonly used terms to describe various treatments. Also check that all treatments are reffered to in a consistant manner: Control treatment is now reffered to as both "CK" and MgSO4.

References and litterature background is fair.

The raw data is shared. However, I do not understand how the raw data in the xls-table of lipid species is used to calculate the data presented in figure 3.

The manuscript contains a well defined and potentially intersting experiment. I do however have some concerns.

Experimental design

The manuscript contains original research that is well within the scope of the journal.

The research question is defined and relevant.

The methods are not well described. In particular it's hard to follow exactly what time points are used and how exactly the plants are inoculated with bacteria (needle less syringe?).

I have several major concerns that needs to be adressed:

1. The methods seems to indicate that a bacterial slurry is defrosted, suspended i Kings B medium and cultivated over night. Thereafter the slurry is diluted in MgSO4 to a low OD and used for inoculation. I worry that there is absolutely no way of knowing that this actually represent bacteria growing in exponential phase rather than mostly dead bacteria after freeze storage. It critically depends on how much of the culture was diluted. The methods seem to indicate that a "tube was just diluted in 5 ml Kings B. In my experience this experiment is usually done on bacteria growing freshly grown on solid medium suspended in MgCl2 och MgSO4.

2. The interpretation of the OPLS-DA plot (Figure 2. . The only comment in the main text is that there is a short distance between samples repesenting systemic leaves from plants inoculated with avirulent and virulent Pseudomonas. However, in fact there is also a short disatnce between these two and local leaf control. In fact the two clusters tha seems to space furthest apart are control systemic leaf local virulent treated. This needs to be better commented on to the very least. To me it seems to indicate that there might be fundamental problems with the experiment.

3. How is the data in the heat plot in figure 3 generated? It is presented as "fold differences", but what is it mormalized to. I cannot understand what it is normalized to as all possible treatments seems to be presented. Additionally, there seems to be quite large variations between the replicates for some of the treatments.

4. It's a bit strange that OPDA-containing galactolipids ("arabidopsides") does not show up in any treatment. Col-0 is well known to make large amounts of these when challenged with an avirulent strain of P. syringae.

Validity of the findings

Provided my aboive concenrns can be met, the findings are valid. However, the final conclusion is not actually supported anyway. Change in the level of a particular metabolite does not imply funtion in SAR. One can safely say that a particular change is associated with the state of SAR induction but nothing can be inferred about actual function.

·

Basic reporting

The manuscript is written clearly throughout, but requires thorough editing to erase minor errors (some are listed below). The introduction into the subject is sufficient, and literature is well referenced (but reference list should be checked again, I notice two missing references, see below. Literature is relevant, only sometimes more care may have been taken to cite the most appropriate (original) publications (see below).
The manuscript is well structured. Figures are of high quality and figure legends are informative. Relevant raw data is provided in the supplement, but could be improved. Specifically, the metabolite table should be supplemented with the respective retention times. QPCR raw data (Ct values) may also be provided as a supplement.

Experimental design

The research question is well defined and relevant. The experimental design (choice of time point, number of replicates, and choice of extraction method) has some limitations (see below) and description of methods is lacking some details (see below). Some phrases used are misleading and should be corrected, e.g. ‘Differentially expressed lipid metabolites (DELM)’ –metabolites are not expressed.
Please discuss your choice of extraction method: which previous publications used this method in plants/with Arabidopsis leaves, which lipid classes were targeted and why?
At which time point were the samples harvested for lipid analysis? This must be mentioned in the methods section, and the choice of this time point should be discussed with respect to previous publications, e.g. those on the timing of known SAR metabolites, such as NHP or SA in systemic leaves.
RT-qPCR: how were the relative expression levels calculated? It looks like different treatments were used as reference (or set to 1) for the different genes.
Line 169: why did you choose different cut-offs for up- or downregulated metabolites? 0.83 vs. 1.2?

Validity of the findings

Generally, a better understanding of the regulation of lipid metabolites during SAR is highly desireable and a welcome addition to existing knowledge of SAR.
The identification of metabolites (Suppl. Table 1) was probably performed in an automated way. Some predictions appear rather unlikely to me, given the previously reported abundance of these metabolites. Some PA levels appear rather high, in some cases these molecules may rather be designated as PE (?). Also PIP signals appear rather high in some cases (?) There are several unlikely predictions e.g. MGDG 29:10, TG 11:4. Hence, manual curation of this list seems to be necessary.
It should be stated (and discussed) that you used rather relaxed criteria for identifying regulated metabolites. With a cut-off of minimum up- or downregulation of 20%. When discussing the relevance of the detected metabolites, you should put the magnitude of regulation in context: How strong were differences in levels of these metabolites in previous publications? Are the magnitudes detected potentially relevant?

You used three biological replicates per treatment. Unfortunately, one of your control treatment samples, SL-CK3, appears to be quite different from the two other control samples (Figure 3). Please discuss. Did this influence the composition of your differentially regulated metabolite list?
To put your results in context, more detail on previous work, specifically: which lipid molecules have been found before in SAR metabolomics approaches, would improve the discussion.
In the discussion, please discuss if you found previously identified SAR-related lipids, and, if not, why? How about Glycerol-3-phosphate, or other molecules such as terpenoids (Attaran, Rostas, Zeier, 2008), sterols (Griebel & Zeier, 2010), tocopherols (Stahl et al., 2019), or the role of lipid peroxidation in SAR responses?
Conclusions are generally well stated, with minor amendments required. If one would be very critical, the title suggests that SAR-associated lipids are identified. Striktly speaking, locally and systemically induced metabolites were identified. If these metabolites play any role in SAR (enhanced resistance of the systemic tissue) is not tested in this study. Therefore, the statement in line 23-24 ‘during SAR’, or line 81 ‘important factor’, ‘essential roles’ are overstatements and should be removed.

Additional comments

51-52 better cite most recent papers on this subject: . Ngou, B. P. M., Ahn, H.-K., Ding, P. & Jones, J. D. G. Nature https://doi.org/10.1038/s41586-021-03315-7 (2021).
54 bacteria-free parts?
58 Li et al. 2020a is not the most appropriate reference, as there are much earlier reports on PIP
80-82 ‘Various PAs, PCs, PEs, and TAGs were identified as important factors in plant immunity, playing essential roles in basal resistance and’ -whether the molecules are essential or not, has to be shown by functional studies
90-91 unclear meaning
104 please give rationale and references for the choice of extraction method
132-133 ‘types of software program’? Specify!
151 planthad
166 metabolites are not espressed. Please rephrase, also your abbreviation ‘DELM’
201 MgSO4-infected?
201-203 expression of GPAT1 is not indicating the importance of PA
206 There are earlier references on the general nature of SAR
231 Fernandez-Delmond et al. not in references
239 Gal?
245 Gao: missing reference
270-283 I do not understand the connection between elevated PE levels and the eut ‘story’ presented in this section...
285 Vir bacteria?
303 this should be expressed more careful and specific
310, 313 TGs were displayed enhanced abundance?
Cfu per mL per leaf disc –better to convert this to the actual area of the leaf disc.
Figure 1: format of column graphs is different in A and C
Figure legend 1: diameter 50 mm?
Figure 3 SL-CK3 appears very different from the two other control samples. D
Figure 5 legend: lipid-related genes?
Figure 5: number of replicates, how were relative levels calculated? Sometimes LL:CK, sometimes SL-CK is set to 1?
Legend to Figure 5: ‘lipid related genes’ ‘treatments with Avr’ These terms are not correct, please specify.

Reviewer 3 ·

Basic reporting

The manuscript is clearly structured and easy to follow. Some sentences need english editing. The introduction with references is suitable. Raw data are supplied.
Description of figures can be improved:

Fig. 2: Would you have not expected that the mock treated local and systemic leaves are in close proximity? Please comment/explain. Did this affect the list of metabolites?

Fig. 3: What do we learn from Fig. 3? There is no further comment on this figure except that this heatmap is there.

Fig. 4: add the observation that the virulent and avirulent strain have more overlap in lipids than local and systemic of the same strain (not surprising but noteworthy)
General: the occurence of several compounds, especially proposed esterified fatty acids, are rather surprising (Suppl. Table 1, Fig, 3, text), for example TG 14:1_10:4_11_4), TG 8:0_11:2_20:5, TG 4:0_18:2_22:6 or TG 6:0_10:3_18:3. Please re-evaluate the data to remove errors generated by automated identification.

Experimental design

The research question is well defined and the data are new. The methods are sufficiently described.

Validity of the findings

Investigations aiming to identify compounds and mechanisms involved in SAR are certainly valuable. However, this manuscript is descriptive resulting in some (new) compounds which show alteration in levels upon pathogen treatment in local and distal leaves. The identification of these metabolites should be reliably correct (see "basic reporting"). Also, the conclusions are rather limited, an overinterpretation should be avoided (see below).
The discussion can be improved:
Especially conclusions that lipids are involved in immune responses (line 291) or even resistance (line 311, 316) cannot be drawn.
paragraph line 250: the conclusion is not clear. Since PCs are present in bacteria, is it possible that the locally detected differencial PCs derive from the bacteria?
For considering a potential relevance of the accumulating compounts please comment the amount and amplitude of change of the metabolites.
Please comment on the timing of responses to Pseudomonas. Which responses are described at which time points and which time point was chosen here? This might have a big impact on the results.
The comparison with published metabolites should be improved. It should be included whether you found an accumulation of lipophilic compounds described by other groups e.g. steroids, tocopherols or oxidized lipids.

Additional comments

specific points:
line 97: bacteria are sedimented are pelleted but not precipitated.
line 164 and legend Fig. 5: MgSO4-inoculated, it is not an infection
Line 166: change the term „differentially expressed lipid metabolites (DELMs)“. Genes are differentially expressed, but not lipids.
line 176: „related to local resistance“: the relation to resistance is not clear.
line 203: PA is an important lipid factor in plant immune processes – unvalid conclusion.

---

## Round 0.2 · Major Revisions

Both reviewers appreciate that your manuscript has been overall improved. However, there are still a few open points and I ask you to address especially the points raised by reviewer #1 in your revision.

·

Basic reporting

The specific points mentioned in the first review have been addressed to a large extent. The manuscript text, especially those passages added in the resubmission, is not ready for publication as language need to be improved and a number of minor errors/typos are present.

The Gao et al. 2020 reference should be mentioned in the introduction, as the data in this study is from the same experiments already described in Gao et al 2020.

Experimental design

not applicable

Validity of the findings

1) Heat map (Figure 3):
Why is one sample missing in the heat map? I believe that the same set of samples should be used for all anlyses.

Compared with the previous version of the heat map, 5 candidates are missing. Why?

There is no explanation how the heat map was generated, what the values represent, and how the shown metabolites were selected.

I have difficulties understanding your answer in the rebuttal letter:
‘We found out from the raw data that some metabolites were not identified in SL-CK-3 group. The expression pattern of SL-CK-3 was different from the other two groups. The result did not influence the composition of our differentially regulated metabolite list because the list of metabolites is made up of 18 samples. "

- I don’t see a causal relationship here? Actually, 17 different metabolites were designated as differentiall abundant in your manuscript.
-
‘If some metabolite is not identified, the value of this metabolite is “0”. ‘

- I don’t understand the argument.

‘Moreover, it may be caused by taking different systemic leaves of locally MgSO4-treated plant (leaves of developmental stages 7-9 as sample).’

Didn’t you use a pool of several leaves for your samples? Please specify this in the methods section.


2) Unlikely metabolite predictions have been successfully removed.

3) The issue with one of the control treatment samples, SL-CK3, has not finally been resolved (see above). Removal from the heat map calculation is not a good option in my opinion. Either this sample is discarded from all analysis or it is included in all. If the sample remains, please mention the impact for the results obtained in the discussion (e.g. differentially regulated metabolite list).

Additional comments

Line 132 Reference missing for the software ‘software program Lipidsearch 4.2.21.’
Line 134-135 using types of software: SIMCA-P14.1.

Line 164-166 All samples in OPLS-DA were within confidence intervals; and this OPLS-DA plot can show the differences in groups. (Figure 2).

Line 173 P19001-Lipidmedian corr).??
Line 174 We found from new heat map ??
Line 107 can refer to
Figure legend to Fig. 3 is difficult to understand: ‘The phylogeny lines on the left represent the clustering analysis results of different samples.’?
Language: -MgSO4 -> MgSO4
-there is positively correlated between PE
-‘normal metabolites’?

Reviewer 3 ·

Basic reporting

The manuscript has overall improved. Some sentences still need english editing.
Description of figures can be improved:
Fig. 2: I still think that it needs to be mentioned that the mock controls (LL-CK versus SL-CK) are so different. Is this due to a wounding response? Or to the use of younger leaves for the systemic response?
Fig. 3 and its description have strongly improved. However, it is still not clear what is shown. The legend says content of lipids. How have the content values between 20 and 35 been generated? Please explain in the legend. Why is the sample SL_CK.3 missing in the figure?
Fig. 4A and B: The number of DLMs is the same as in the last version even though the number of metabolites was reduced from 173 to 127. For example TG (5:0_18:2_18:3) was removed but the number of DLMs mentioned in the text is still the same. Please re-check this figure and the corresponding paragraph.

Experimental design

no comment

Validity of the findings

The discussion has improved.
Conclusions: line 342: this was not changed compared to the last version even though the 20 and 13 DLMs refer to Table 2 which only shows a total of 18 metabolites. Please re-check.

Additional comments

line 174: remove „new“
lines 181, 342 and 346: change DELM to DLM

---

## Round 0.3 · Minor Revisions

Thank you for your revisions that greatly improved the manuscript! You carefully addressed all concerns from the previous reviews. However, both reviewers and I suggest during both rounds another round of careful editing. Sometimes the sentence structure reads a bit difficult, the sentences are incomplete or strange phrasing was used (e.g. the Abstract, Conclusion, and the Methods sections). This can be either done by yourself, a proficient English-speaking colleague, or a proofing service.

·

Basic reporting

Thank you for your revisions. The revised manuscript has been greatly improved, and carefully addressed my concerns from the previous review. I thank the authors for their detailed responses to my comments. The updated manuscript with the new Figure 2 (OPLS-DA), Figure3 (heat map), Figure4 (venn) and differentially regulated metabolite list (table 2) is much clearer. I think this is an interesting publication and the paper will be a useful reference.
I still spotted some occasional typographic errors (e.g. MgSO4 in figure legend to Fig.5.), and the term ‘Differentially lipid metabolites (DLM)’ maybe should be changed into ‘differentially regulated metabolites’, but I am not a native speaker, and I believe this will be taken care of when checking the proofs. The somewhat low resolution of Figure 2 and Figure 4 in the pdf file seems to be not present in the original file. I assume this will be taken care of by the production department.

Experimental design

OK

Validity of the findings

OK

Reviewer 3 ·

Basic reporting

I do not have any further comments

Experimental design

I do not have any further comments

Validity of the findings

I do not have any further comments

Additional comments

I do not have any further comments

---

## Round 0.4 · Minor Revisions

Thank you very much again for this careful revision. However, I have to admit that the text still harbors a number of spelling errors and is still in some cases hard to understand, e.g. the newly added sentences at the end of the abstract need to be rephrased and DLM should be replaced by DRM since you changed it into differentially regulated metabolites. Therefore I strongly encourage you to seek the help of a proficient English speaker or a professional service for language editing of your otherwise very interesting manuscript.

---

## Round 0.5 · accepted · Accept

Thank you very much for this very careful revision!